# Longitudinal links between behavioral activation coping strategies and depressive symptoms of U.S. adults living alone during the COVID-19 pandemic

**Seoyoun Kim**[1]*, **Hyunwoo Yoon**[2], **Patricia Morton**[3], **Yuri Jang**[4]

**1** Department of Sociology, Texas State University, San Marcos, Texas, United States of America,
**2** Department of Social Welfare, Kongju National University, Gongju, South Korea, **3** Department of
Sociology, Department of Public Health, Wayne State University, Detroit, Michigan, United States of America,
**4** Suzanne Dworak-Peck School of Social Work Edward R. Roybal Institute on Aging, University of Southern
California, Los Angeles, California, United States of America

* skim182@txstate.edu

Longitudinal links between behavioral activation
coping strategies and depressive symptoms of U.
S. adults living alone during the COVID-19
pandemic. PLoS ONE 17(5): e0267948. https://doi.
org/10.1371/journal.pone.0267948

University Purdue University at Indianapolis,
UNITED STATES

**Data Availability Statement:** Data are available for
download for any registered UAS data user.
Researchers should fill out a data request form

## Abstract

The COVID-19 pandemic and related physical distancing measures have posed a significant threat to the mental health of adults, particularly those living alone. Accordingly, the World Health Organization implemented the #HealthyAtHome program, encouraging people to keep in regular contact with loved ones, stay physically active, and keep a regular routine. The current study aims to examine a micro-longitudinal link between behavioral activation coping strategies (exercise, meditation, relaxation, and social connection) and depressive symptoms among adults who lived alone during the COVID-19 pandemic. We used 21 biweekly waves of longitudinal data from the Understanding America Study (UAS) collected between April 2020 and February 2021 (N = 1,280). The multilevel models with correlated random effects were estimated to examine lagged effects of coping strategies (t-1) on depressive symptoms (t). The results showed that exercise was predictive of lower depressive symptoms even after controlling for time-invariant and time-varying covariates. The results showed that modifiable lifestyle factors, such as taking time to exercise, may be beneficial for the mental health of Americans living in single-person households.

## Introduction

The COVID-19 pandemic has presented unique public health challenges to everyone. In particular, various physical distancing measures (e.g., a distance of 6-feet from others, shelter-in-place order) to alleviate the spread of COVID-19, though necessary, can result in social and physical isolation, which subsequently contributes to stress, anxiety, and depression. Previous studies after the SARS, MERS, and during COVID-19 pandemics show psychological ramifications of physical and social isolation, including depressive symptoms, anxiety, insomnia, loneliness, obsessions, delirium, emotional disturbance (e.g., self-blame, guilt), and even post-traumatic stress symptoms [1–4].

describing their analysis plan before downloading the data. Once they have filled out the form, they will have access to download the data from the data pages. https://uasdata.usc.edu/index.php.

**Funding:** The authors received no specific funding for this work.

**Competing interests:** The authors have declared that no competing interests exist.

Among the vulnerable groups affected by the COVID-19, individuals living alone may be particularly at risk for mental problems caused by physically distancing and social isolation. An estimated 34.8 million single-person households in the United States [5] face adverse mental health outcomes potentially due to economic disruptions such as a loss of income, limited access to usual sources of social and emotional support and/or regular social interactions at a time of crisis. Lack of cohabitants also means that those living alone are faced with negative emotions, including loneliness and isolation although living alone doesn't always mean the lack of active social lives. In fact, a recent U.S. census report documented that, between October and November of 2020, 31% of adults living alone reported depression, compared to 26% of people living with cohabitants [5]. One study showed that during the COVID-19 outbreak, people living alone often felt scared, isolated, and deprived of their usual interactions [6]. Therefore, it is important to understand the mental health issues surrounding COVID-19 and identify preventative coping strategies for those living alone to alleviate mental health problems caused by the COVID-19 pandemic for this uniquely vulnerable group.

When faced with this a macro-level stressor like a pandemic and related negative emotion, coping strategies may lower adverse mental health outcomes. Coping is defined as "the thoughts and behaviors used to manage the demands of situations that are appraised as stressful" [7] (p. 745). What sets apart the COVID-19 pandemic from other infectious diseases such as SARS, Ebola, or Swine Flu, is the physical and social isolation. This sudden rupture in social relationships can be compensated, at least partly, by carrying out various activities that are not dependent upon face-to-face interactions including social interaction via social media, self-care activities, and physical activities. These activities, called 'behavioral activation,' are robust predictors of improved psychological well-being after a national tragedy like 9/11 [8] and commonly documented in interventions for depressive disorders [9]. Specifically, those employing behavioral activation are able to cultivate positive emotions, bounce back easily from negative emotions, and have psychological ability to manage daily stressors [10]. Some exploratory studies during the COVID-19 pandemic examined exercise, maintaining social connections, cooking, learning new skills, and limiting consumptions of news about the outbreak as potentially protective against depressive symptoms [11–14]. Though there are no defined categories of behavioral activation coping strategies, in the current study, we focus on four common strategies: exercise, relaxation, meditation, and social connections.

Specific to COVID-19, engaging in frequent physical activity has shown to be associated with better mental and physical health during physical distancing, regardless of age or geographic regions [15–18]. For example, Schuch and colleagues (2020) showed that more than 30 minutes of moderate to vigorous physical activity was associated with 30% lower likelihood of experiencing depression or anxiety [11]. A recent systematic review of the literature also reports that higher time spent in moderate physical activity was associated with 12–32% lower likelihood of exhibiting depressive symptoms and 15–34% lower probability of presenting anxiety [18]. Meditation or relaxation may also be protective against the isolating effects of physical distancing. A study of 1,021 U.S. adults showed that actively taking time to relax and time away from the news were protective against depressive symptoms during the pandemic [14]. Another study commented that mindfulness meditation corresponded to significantly lower mental discomfort and distress by cultivating the ability to sit with circumstances and observe our thoughts without judgements [19]. A recent rapid systematic review revealed that two randomized-control trials (RCT) of mindfulness-based interventions were effective in reducing loneliness [20]. One major strength of relaxation and meditation programs is that they can easily be adapted via online programs or apps for individuals living alone. Finally, seeking social connections through phone calls, emails, and video calls may be a direct remedy against social isolation. One study proposed that high quality social connections during the pandemic

enhance sleep quality and reduces depressive symptoms, loneliness, anxiety, and irritability [21]. Several RCTs showed that videoconferencing programs and peer interactions were effective in reducing loneliness and depressive symptoms among older adults and nursing home residents [22]. However, some studies did not show significant effects of social networking interventions on depressive symptoms [23].

Finally, it is worth noting that there are some avoidant and less adaptive coping strategies. For instance, smoking, drinking, and using recreational drugs during the pandemic were predictive of depression, anxiety, and insomnia [24]. Though substance use may be counterproductive to the benefits of exercise, relaxation, meditation, and social connections, it would be informative to determine whether the impact of the healthful behavioral activation coping would sustain regardless of the presence of the maladaptive coping strategies.

In order to alleviate the mental health consequences of physical distancing measures, the World Health Organization (WHO) implemented the #HealthyAtHome program, encouraging people to adopt various preventive coping strategies such as keeping in regular contact with loved ones, staying physically active, and keeping a daily routine. In addition, the WHO recommends to increase social connection through virtual media within families and communities [25]. Though recent studies have documented and proposed positive coping strategies in a matter of months, a major caveat of the current literature is that many of them are cross-sectional. Further, no study to date focuses specifically on both those living alone, one of the most vulnerable populations for social isolation, and ascertaining the links between coping strategies and depressive symptoms over an extended period of time.

Given the heightened risks of social isolation and mental problems among those living alone as a result of the COVID-19 pandemic, there is an urgent need to identify positive coping strategies to help develop and tailor effective public health interventions. The current study aims to examine a micro-longitudinal link between behavioral activation coping and depressive symptoms among those living alone during the COVID-19 pandemic in a population-based survey of the U.S. adults. Building upon the aforementioned literature, we hypothesized that four types of behavioral activation coping strategies (exercise, meditation, relaxation, and social connection) would be associated with lower levels of depressive symptoms for those living alone.

## Methods

### Data

To address the study objectives, we analyzed data from 21 biweekly waves of the Understanding America Study (UAS). The UAS is nationally representative, an ongoing panel study of community-dwelling adults. The data collection occurred every two weeks. The current study used data from 21 waves of the UAS surveys specifically tracking the psychological effects of the COVID-19 pandemic from April 1st, 2020 to February 3rd, 2021. The correlation of living arrangement (living alone vs. living with others) ranged from 91–99% across the 21 waves. Therefore, from the 8,547 UAS eligible respondents, 1,321 respondents who lived alone at any 2-week period during the study period were included in the analyses. 41 cases (3%) were dropped due to missing data in independent variables (i.e., coping strategies), resulting in a final analytic sample size of 1,280. The missing data are very small. Moreover, the cases with missing values (N = 41) and the 1,280 individuals in the sample were not significantly different in terms of age, depressive symptoms, education, income, chronic conditions, and substance use. Thus, we employed listwise deletion technique.

## Measures

**Behavioral activation coping strategies.**   *At each wave*, *respondents* were asked how many days in the past week they have engaged in the following activities: exercise, meditation, taking time to relax, and connecting socially with friends and family (either online or in-person). Each item ranged from 0 to 7 days.

**Depressive symptoms.**   At each wave, depressive symptoms were measured using the Patient Health Questionnaire-4 (PHQ-4), addressing the four core symptoms of depression. Respondents were asked how often in the past 2 weeks they have experienced 1) feeling nervous, anxious, or on edge, 2) feeling down, depressed, or hopeless, 3) not being able to stop worrying, and 4) having little interest or pleasure in doing things. Each response ranged from not at all (0), several days (1), more than half the days (2), and nearly every day (3). The total scores across all 4 items ranged from 0 to 12. Per PHQ-4 recommendations, total scores were grouped into normal (0–2), mild (3–5), moderate (6–8), and severe (9–12) depressive symptoms [26].

**Covariates.**   The analyses controlled for time-invariant and time-varying covariates relevant for depressive symptoms and/or coping strategies. Age was coded in years, and sex was coded as female (0) and male (1). Race was categorized into non-Hispanic White (0), non-Hispanic Black (1), and other race (2), including American Indian, Alaska Native, Asian, Hawaiian, Latinx, Pacific Islander, and multiracial individuals. They were combined into a single group due to the small sample size. Current marital status was classified as married (0), separated/divorced (1), widowed (2), and never married (3). Education was measured as less than high school (0), high school (1), some college (2), and bachelor's degree or more (3). Household income was categorized into less than $50,000 (0), $50,000 to less than $100,000 (1), and $100,000 or more (2). Chronic conditions were created by totaling the number of 8 common chronic symptoms, including diabetes, cancer, heart disease, hypertension, asthma, COPD, kidney disease, and autoimmune disorder. Age, race, marital status, education, income, and chronic conditions were included as time-invariant covariates. Finally, substance use was measured by asking respondents how many days of the week they smoke, drink alcohol, or use recreational drugs like marijuana. Each question ranged from 0 to 7 days. The substance use question was obtained by averaging the three questions (smoking, alcohol use, and creational drugs). This variable was included as time-varying covariates in order to account for changes in substance use.

## Statistical analysis

Descriptive statistics were conducted to examine how depressive symptoms and independent variables have changed between April 2020 and February 2021. All covariates except for substance use were considered time-invariant since they did not show significant changes over the study period. Then, the multilevel models with correlated random effects were estimated to address lagged effects of coping strategies (t-1) on depressive symptoms two weeks later (t). The analysis used the *xthybrid* command with clustered standard errors, fixed effects, and correlated random effects. This model is preferred to standard random-effects or fixed-effects models since it does not assume normal distributions of the dependent variable or unobserved effects. Further, this model controls for between-individual effects (i.e., the possibility that coping strategies are inherently different among people with and without depressive symptoms) when estimating within-individual effects. Thus, significant within-person estimates would provide better inference [27, 28]. Finally, all statistical analyses include sampling weights created by the UAS in order to produce less biased population estimates. The sampling weights

were generated at each wave and adjust for unequal sampling probabilities of participants into the UAS survey [29].

Several sensitivity analyses were conducted to test the robustness of the final model. First, coping strategies were tested with multiple specifications. We created four other versions indicating whether 1) the respondent engaged in a given activity any day of the week; 2) the respondent engaged in a given activity more than 3 days of the week; 3) the number of total activities partaken any day of the week (0–4) and; 4) the number of activities participated more than three days a week (0–4). The substantive conclusions or patterns did not change. Therefore, the final analyses used the original coding of 0–7 days. Second, other covariates, such as region of the country, Supplemental Nutrition Assistance Program (SNAP) eligibility, employment status, and employment as an essential worker were considered; however, they were neither significant nor changed the substantive conclusions of the analyses and subsequently removed from the final analyses for a more parsimonious model. All analyses were conducted using Stata 17.

## Results

Table 1 presents the descriptive statistics of the core study variables. Respondents were on average 55.96 years old. 39% were male, 75% were non-Hispanic White, 12% were non-Hispanic Black, and about 14% were in other racial groups. In terms of marital status, 6% were married, 40% were divorced, 15% were widowed, and 39% were never married. A majority of the respondents hold a bachelor's degree or more (44.7%), which is higher than the U.S. population average of 38%. 52% of the respondents were employed. Average income level was less than $50,000. The average number of chronic conditions was 1.02. At Wave 1, 17.85% of the respondents exhibited moderate to severe depressive symptoms, compared to 11.19% at the last wave. As graphically represented in Fig 1, depressive symptoms were quite high on average at the beginning of the study (April 1, 2020) but drastically decreased in the next few waves, consistent with other studies [30]. On average, respondents reported 3.7 days of exercise, 2 days of meditation, 5.4 days of relaxation, and 4.7 days of socially connecting with someone at Wave 1. By Wave 21, the depression level was lower than the first wave (0.48). Respondents also reported slightly fewer days of engaging in various coping activities (exercise = 3.3; meditation = 1.9; relaxation = 4.6; social connection = 4.5).

Table 2 summarizes the results from the longitudinal relationship between coping strategies and depressive symptoms using a multilevel random-effects model. In Model 1, including only coping strategies, those who engaged in frequent physical activities were likely to report lower levels of depressive symptoms (ß = -0.07, $p<.001$). This effect remained in Model 2 when basic demographic variables were included. In Model 3 with the full set of covariates, exercise remained a significant predictor of depressive symptoms two weeks later (ß = -0.08, $p<.001$). More frequent exercise was associated with lower levels of depressive symptoms in two weeks, even after controlling for demographic covariates, chronic conditions, and depressive symptoms at the previous wave. More specifically, one standard deviation increase in exercise was associated with 0.08 standard deviation lower depressive symptoms. The size of this coefficient was similar to that of income (ß = -0.09). On the contrary, meditation, relaxation, or social connection were not associated with depressive symptoms. Model 3 yielded the best model fit indices (AIC = 26249.6; BIC = 26466.6) compared to the first two models. The tests of the random-effects assumption showed the preference for the random-effects model except for social connection (exercise $p = .0004$, meditation $p = .0008$, relaxation $p = .0001$, social connection $p = .0918$).

**Table 1. Descriptive statistics of the study variables (n = 1,280).**

|  | Mean (SD) or % |
|---|---|
| Depressive Symptoms, W1 | |
| None | 57.91% |
| Mild | 24.24% |
| Moderate | 10.85% |
| Severe | 7.00% |
| Behavioral Activation Coping Strategies, W1 | |
| Exercise | 3.69 (2.33) |
| Meditation | 2.01 (2.70) |
| Relaxation | 5.44 (2.23) |
| Social Connection | 4.73 (2.56) |
| Age | 55.96 (16.22) |
| Sex | |
| Female (ref) | 60.67% |
| Male | 39.33% |
| Race | |
| White (ref) | 74.66% |
| Black | 11.81% |
| Other Race | 13.53% |
| Marital Status | |
| Married (ref) | 6.34% |
| Divorced | 39.59% |
| Widowed | 14.62% |
| Never Married | 39.45% |
| Employed | 52.52% |
| Income | |
| <$50,000 (ref) | 63.29% |
| $50,000- <$100,000 | 24.93% |
| $100,000 or more | 11.77% |
| Education | |
| Less than high school (ref) | 4.24% |
| High school | 38.38% |
| Some college | 12.67% |
| Bachelor's degree or higher | 44.70% |
| Chronic Conditions | 1.02 (1.15) |
| Substance Use | 0.76 (1.09) |

## Discussion

In addition to the macro-stress of a pandemic, public health measures to curtail the spread of COVID-19 may inadvertently compound the mental health consequences of COVID-19. Though necessary, physical distancing may be particularly detrimental to individuals residing alone. The present study sought to identify coping strategies that are associated with lower depressive symptoms among single-person U.S. households during the pandemic. Among the four behavioral activation coping strategies, exercise was a consistent and robust predictor of depressive symptoms, even after controlling for several mental health risk factors, time-varying and time invariant covariates, and random effects. Exercise remained a significant predictor of lower depressive symptoms even when substance use was introduced in the final model.

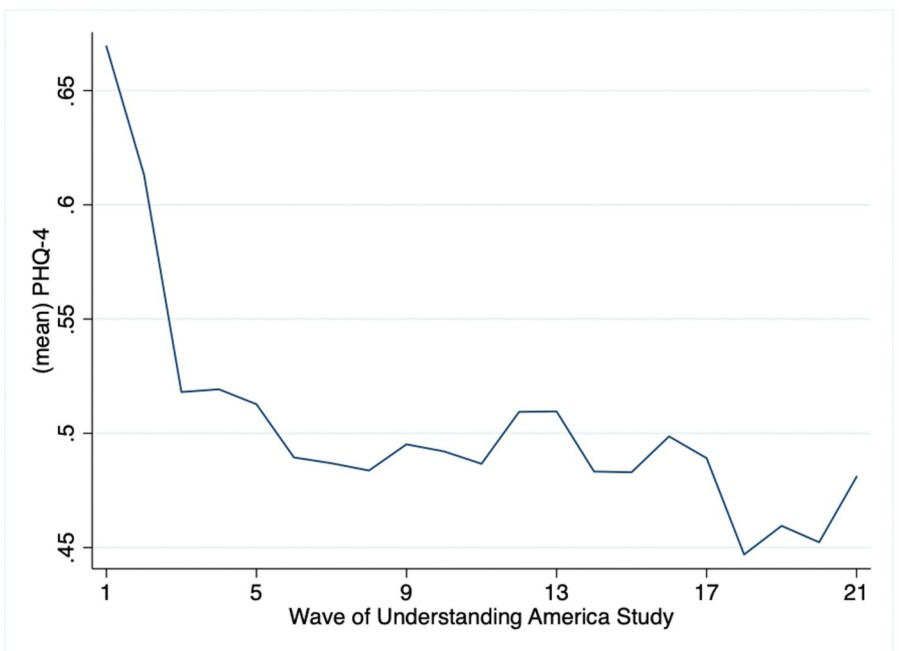

**Fig 1. Changes in depressive symptoms, April 2020-February 20.**

Substance use was associated with higher levels of depressive symptoms [24], while exercise remained predictive of lower depressive symptoms.

There are several plausible explanations for the mental health benefits of exercise for both those living with cohabitants and those living alone. In general, exercise has immediate and long term psychological, cognitive, and physiological effects that can reduce depressive symptoms, particularly in the context of a pandemic [18, 31]. Exercise can regulate physiological mechanisms, such as exercise-induced increase in blood circulation to the brain, sympathetic nervous system, immune system, and hypothalamic-pituitary-adrenal (HPA) axis, which control a range of stress hormones, thereby enhancing psychological well-being [32]. Indeed, the well-documented beneficial effects of exercise on depressive symptoms may explain why exercise was a salient predictor of lower depressive symptoms during the pandemic in single-person households. For those living alone, quarantine and physical distancing may have led to elevated levels of social isolation and loneliness, which have been identified as independent risk factors for physical inactivity [33]. The benefits of exercise on depressive symptoms were well-documented in other population-based studies and was again a salient predictor of depressive symptoms during the pandemic in single-person households. The findings of the current study suggest physical exercise as potentially a non-pharmacological intervention for mental health in the face of physical distancing during the COVID-19 pandemic.

Intriguingly, the other potential coping mechanisms were not associated with depressive symptoms among people residing alone. In particular, the non-significant finding of social connection with friends and family was counterintuitive (yet in line with some studies [23]) given the benefits of social relationships and increased social isolation amidst the pandemic. There may be multiple reasons for the lack of associations. First, it is possible that meditation, relaxation, and social connection are effective for different types of mental health outcomes such as anxiety. For example, frequent meditation and relaxation could be associated with anxiety via activating parasympathetic nervous system (e.g., slower heart rate) [34]. Second, social

**Table 2. Two-level random effects model (n = 1,280).**

| | Model 1 | | Model 2 | | Model 3 | |
|---|---|---|---|---|---|---|
| | ß | SE | ß | SE | ß | SE |
| Behavioral Activation Coping Strategies | | | | | | |
| Exercise | -0.07*** | .02 | -0.07*** | .01 | -0.08*** | .01 |
| Meditation | 0.03 | .01 | 0.03 | .01 | 0.02 | .01 |
| Relaxation | -0.01 | .01 | -0.01 | .01 | -0.01 | .01 |
| Social Connection | -0.01 | .01 | 0.00 | .01 | 0.01 | .01 |
| Covariates | | | | | | |
| Age | | | -0.03*** | .01 | -0.05*** | .01 |
| Male | | | -0.88*** | .19 | -0.96*** | .19 |
| Black | | | -1.78*** | .28 | -1.86*** | .28 |
| Other Race | | | 0.30 | .26 | 0.35 | .26 |
| Divorced | | | -0.13 | .37 | -0.30 | .37 |
| Widowed | | | -0.47 | .43 | -0.60 | .42 |
| Never Married | | | 0.20 | .37 | 0.10 | .37 |
| Employed | | | | | -0.67** | .22 |
| Income | | | | | -0.09** | .03 |
| Education | | | | | 0.24* | .10 |
| Chronic Conditions | | | | | 0.41 | .09 |
| Substance Use | | | | | 0.66*** | .09 |
| AIC | 26692.6 | | 26446.9 | | 26249.6 | |
| BIC | 26805.2 | | 26615.6 | | 26466.6 | |

*$p<.05$;

**$p<.01$;

***$p<.001$

connection in particular is a complex concept encompassing the frequency and the quality of human interactions. Connecting with family and friends can be positive due to perceived support, quality interactions, and even commiseration; it can be negative due to lack of understanding, bickering, or unsolicited advice or help. A recent study also shows that virtual social connections can promote negative health outcomes (e.g., Zoom fatigue) [35]. Therefore, future studies should include more complex measures of social connection in predicting depressive symptoms. Indeed, the function of various types of behavioral activation coping strategies should be revisited with multi-item scales with good psychometric properties.

The findings herein offer several implications for public health. First, these findings identify a relatively low-cost and low-risk public health intervention to combat mental health issues during the pandemic for those living alone. Sedentary behaviors have increased during the pandemic [36], so it is possible that some level of voluntary physical exertion, in tandem with the psychological, cognitive, and physiological benefits, may be more beneficial given the unique circumstances. The Centers for the Disease Control and Prevention (CDC) should take a more proactive approach to encourage physical activity during the ongoing, significant COVID-19 public health crisis. For example, the CDC could expand its Move Your Way program and How to Be Physically Active While Social Distancing webpage to include an array of suggested physical activities. These activities should be suited for individuals with a variety of physical abilities and be able to be done during gym closures. For single-person households,

group classes could be offered to promote physical activity and social connection. The WHO #HealthyAtHome can provide a template for disseminating evidence-based research on physical activity to the public. Second, these findings highlight the preventative potential of exercise to curtail mental health issues during large-scale public health threats that require physical isolation measures. In the face of a future macro-stressor requiring physical isolation, exercise may be an effective and low-cost public health strategy to implement early on. Non-pharmacologic public health strategies like a physical activity initiative during similar large-scale events may be particularly useful to inhibit disturbances on mental health in countries like the U.S. that do not have strong healthcare infrastructure.

## Strengths and limitations

Our study has a number of strengths. One of the main contributions to the literature is that this study uses a prospective design with repeated measures and a large, national sample of US adults. Further, the data were collected during the pandemic instead of afterwards. To the authors' knowledge, this is the first study to focus on coping strategies and mental health of U. S. adults living alone during the pandemic. Although most COVID-19 research has focused on mental health consequences of social isolation, the present study sought to understand the well-being of another high-risk group during the pandemic: adults living alone. In addition, the modeling approach adjusted for time-invariant and time-varying covariates to adjust for confounding and provide better inference about changes. The findings provided evidence that exercise is associated with lower depressive symptoms two weeks later, although we cannot rule out the potential reverse causality. Finally, the findings inform not only the current challenges emerging from the COVID-19 pandemic, but also future public crises with similar physical distancing measures to protect the mental health of groups who reside alone or experience physical isolation.

Some study limitations should be noted. Though the analytical technique used in the study addresses endogeneity due to omitted variables [28] it does not deal with reverse causality between measures of depressive symptoms and coping strategies. We conducted supplementary analyses using depressive symptoms as predictors and found that depressive symptoms are not significantly associated with the four coping strategies two weeks later. This does not mean that we can rule out reverse causality but indicates that during the study period, coping strategies were predictive of depressive symptoms and *not vice versa*.

Also, there may be covariates not accounted for in the analysis, such as built environment, state policies, social norms, personality traits, and genetic predispositions. Although the current dataset does not include these potential factors, further research should consider these influences. Whereas the current study sought to identify coping strategies to enhance mental health, it did not examine the mechanisms linking exercise to depressive symptoms nor did it explore the context of exercise due to data limitations. Future projects should identify the potential pathways linking exercise and mental health and explore the context of exercise, such as type (aerobic vs. resistance, muscle-strengthening exercise), intensity and duration of exercise, virtuality (in-person vs. online), environment (indoor vs. outdoor), and sociability (individual vs. group activities). Finally, measures of coping strategies do not capture the complexity of the concepts. For example, a measure of meditation could ask the type (e.g., mindfulness, contemplative, Buddhist), duration, and timing. Measures of social connection may ask the quality of the interaction, network size, mode of communication, and the type of relationship (partner vs. family vs. friends). Further research should include detailed items of coping strategies in predicting mental health outcomes.

## Conclusion

The current study finds that physical exercise may be an effective mental health intervention in order to combat the potentially negative effects of physical distancing measures. Epidemiologists are uniquely positioned to implement policies for the target population and advance the understanding of mental health outcomes in the midst of the pandemic. Interdisciplinary efforts between epidemiologists, public health scholars, policymakers, and clinicians will be invaluable in designing interventions to improve health outcomes during public health crises.

## Author Contributions

**Conceptualization:** Seoyoun Kim.

**Formal analysis:** Seoyoun Kim.

**Methodology:** Seoyoun Kim.

**Writing – original draft:** Seoyoun Kim, Hyunwoo Yoon, Patricia Morton, Yuri Jang.

**Writing – review & editing:** Seoyoun Kim, Hyunwoo Yoon, Patricia Morton, Yuri Jang.

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
