## [Decision Letter · Decision Letter 0]

10 Dec 2021

PONE-D-21-32992Longitudinal Effects of Behavioral Activation Coping Strategies on Depressive Symptoms of U.S. Adults Living Alone during the COVID-19 PandemicPLOS ONE

Dear Dr. Kim,

Thank you for submitting your manuscript to PLOS ONE. After careful consideration, we feel that it has merit but does not fully meet PLOS ONE’s publication criteria as it currently stands. Therefore, we invite you to submit a revised version of the manuscript that addresses the points raised during the review process. Typically, two reviewers would review a manuscript before an editorial decision; however, given the difficulties finding reviewers and the need for timely decisions, I have elected to move forward with my decision based on Reviewer 1's comments. Reviewer 1 has provided very thorough and detailed feedback on how to improve the manuscript and meet the PLOS ONE publication criteria. In particular, Reviewer 1 highlights some methodological concerns including whether the data are representative, the role of unobserved confounders, and the use of causal language in the context of the current research design. I share Reviewer 1's concerns, and I ask that you do your best to address their comments.

We look forward to receiving your revised manuscript.

Kind regards,

Kenzie Latham-Mintus, PhD, FGSA

Academic Editor

PLOS ONE

Journal Requirements:

We note that you have indicated that data from this study are available upon request. PLOS only allows data to be available upon request if there are legal or ethical restrictions on sharing data publicly. For information on unacceptable data access restrictions, please see http://journals.plos.org/plosone/s/data-availability#loc-unacceptable-data-access-restrictions. 

Reviewers' comments:

Reviewer's Responses to Questions

**Comments to the Author**

1. Is the manuscript technically sound, and do the data support the conclusions?

Reviewer #1: Partly

2. Has the statistical analysis been performed appropriately and rigorously? 

Reviewer #1: No

3. Have the authors made all data underlying the findings in their manuscript fully available?

Reviewer #1: Yes

4. Is the manuscript presented in an intelligible fashion and written in standard English?

Reviewer #1: Yes

5. Review Comments to the Author

Reviewer #1: The paper aims to test if behavioral activation coping strategies reduce depressive symptoms during COVID-19 period for adults living alone. Because COVID-19 pandemic had substantial effects on psychological well-beings for adults in the U.S., empirical evidence for the effect of coping strategies on depressive symptoms would tell us how adults in the U.S. can deal with this unusual situation. I also appreciate their focus on those living alone, a potentially vulnerable population. Below are my comments to this study which the authors may have to consider:

PLOS ONE's criteria: Experiments, statistics, and other analyses are performed to a high technical standard and are described in sufficient detail.

There are several methodological concerns in the current submission. First, although the authors described that the Understanding America Study (UAS) is a nationally representative survey, there is an implausible number in descriptive statistics. Looking at Table 1, among the analytical sample in the current study, 44.12% of respondents hold graduate degree. Given that the percentage of those who have bachelors or higher (not a graduate degree) in Massachusetts (one of the states with the highest proportion of college graduates in the U.S.) is 44%, I suspect that this is too high to consider the dataset is nationally representative. I appreciate it if the authors go back to the data again to confirm whether you do not have any coding mistakes for educational attainment. Relevantly, I could not find whether the authors used appropriate weights to account for the probability to participate in the survey. Because UAS provides sampling weights, they should incorporate them in the analyses.

Second, the methodology in this paper cannot make causal claims, though the goal of this paper is very causal (“we hypothesized that four types of behavioral activation coping … would lower the levels of depressive symptoms for those living alone”). Although I recognized that the authors tried to account for time-variant by adding control variables, this is not enough because there could (and should) be unobserved confounders. This is an important limitation in this paper because the paper aims to test whether coping strategies reduces depressive symptom and the authors provided policy implications by assuming that the observed correlation is causation.

Here is one example scenario that the observed correlation is not causation. The paper tried to examine the relationship between lagged coping strategies in t-1 and depressive symptom in t; however, the authors did not account for the possibility that the lagged coping strategies in t-1 may be affected by depressive symptom in t-2. This induces a selection issue: less depressed individuals in t-2 are more likely to exercise in t-1; therefore, the authors observed the negative association between exercise in t-1 and depression in t. There are a lot of other scenarios that would show that the analytical strategy in this paper cannot make causal claims. The authors should draw DAG to show their (assumed) data generating process, and explain why their assumption is valid. Alternatively, the authors may be able to use established causal inference techniques to make compelling causal claims. Otherwise, I appreciate it if the authors extensively revise the manuscript to avoid causal languages, including their hypothesis.

Third, the sample size is not consistent across the model. In the descriptive statistics, the sample size is 1,321, but the analytical sample in Table 2 is 1,280. The authors should use the same analytical sample.

Fourth, I am not sure if listwise deletion is an appropriate approach to treat the missing values in independent variables, though the number of respondents with missing values is small. Probably, it is worth to try multiple-imputation for the robustness check. Furthermore, they explicated that they dropped observations if respondents do not have values for coping strategies, but I would like to hear whether there are no missing values in the other independent variables.

Fifth, I am somewhat concerned about the very high PHQ-4 score in Wave I, but there was no such increase afterward. I do understand that this is the first time that COVID-19 cases have drastically increased in the U.S., but why couldn't we see a similar increase afterward. Is it expected? Could you see a similar trend for those not living alone? Are there any studies which found similar results?

Finally, I am interested in why the authors included time-invariant variables even though they incorporated fixed effects which should account for time-invariant effects.

PLOS ONE's criteria: Conclusions are presented in an appropriate fashion and are supported by the data

Overall, the discussion section sounds fine. However, I have several points that I would like the authors to incorporate. First, I would like the authors to add how substantially important exercise is. I understood that exercise is negatively associated with depressive symptoms among those living alone, but the authors did not provide interpretations for the size of coefficients. This discussion will strengthen the argument of policy interventions.

I also would like the authors to reconsider the interpretations of non-significant associations between some coping behaviors (i.e., meditation, relaxation, and social connection) and depressive symptoms. The authors speculated that increased sedentary behaviors during the COVID-19 increases the importance of exercise, but this should not be the reason why the authors did not see significant association between the other coping behaviors. Specifically, I do not think that social connection is not associated with depressive symptoms because of the increased seating time.

Additionally, I am not sure whether we can generalize these results to another large-scale traumatic event (“Second, these findings … a large-scale traumatic event”). Because COVID-19 pandemic carries unique features (like social distancing), this may be different from, for example, September-11 attacks. I appreciate it if I can hear the justification why the authors can conclude in this way.

Further, I did not understand this sentence: “the present study sought to understand how the pandemic influenced the well-being of another high-risk group: adults living alone.” This study does not test the effect of COVID-19 pandemic on depressive symptoms, but examined the association between coping strategies and depressive symptoms during COVID-19 pandemic.

Last but not the least, given the above-mentioned causality issue, I appreciate it if the authors avoid causal languages and add this issue as a limitation in the discussion section.

PLOS ONE's criteria: The article is presented in an intelligible fashion and is written in standard English

In general, the article is presented in an intelligible fashion and is written in standard English. However, as noted in this memo multiple times, the submission is mixing correlation and causation. I could see many causal languages, such as “effect” and “influence,” in the main text. Again, the authors should avoid these causal languages unless they can identify causal effects by using established causal inference techniques.

Another thing that I was concerned about is “general population” in the discussion section. The authors compared those living alone to the general population, but I do not understand what the “general population” means. Are those not living alone general population? If so, why can we consider that those living alone are not “general population?”

Very minor points: (1) probably, the authors should avoid abbreviations, such as “SNAP eligibility,” (2) probably, the reference #26 is wrong because I found PHQ-15 and PHQ-5 but not PHQ-4, (3) the authors should use a consistent language (e.g., COVID-19 pandemic or COVID-19 epidemic).

6. PLOS authors have the option to publish the peer review history of their article (what does this mean?). If published, this will include your full peer review and any attached files.

Reviewer #1: No

---

## [Author Response · Author response to Decision Letter 0]

7 Mar 2022

Longitudinal Links between Behavioral Activation Coping Strategies and Depressive Symptoms of U.S. Adults Living Alone during the COVID-19 Pandemic

PONE-D-21-32992

Dear Editor Kenzie Latham-Mintus, 

We appreciate the careful reading of our paper and constructive comments by the PLOS One editor and reviewer. The reviewer and the editor agreed that the paper examines an important and timely research question; however, they also noted several areas of improvement, which we have addressed in this letter and the revised manuscript. As highlighted in your letter, we have addressed concerns related to the representativeness of the data (see comment 1), unobserved confounders (see comment 2), and the use of causal language (see comments 2, 11, and 12). 

We addressed all issues indicated in the reviewer report and highlighted the changes in the manuscript using red font. Although not detailed in the memorandum, we also deleted elements of the original text and reference list. We list the line number for each revision. Reviewer comments are in all caps; our response uses traditional capitalization. 

Reviewer 1.

1. FIRST, ALTHOUGH THE AUTHORS DESCRIBED THAT THE UNDERSTANDING AMERICA STUDY (UAS) IS A NATIONALLY REPRESENTATIVE SURVEY, THERE IS AN IMPLAUSIBLE NUMBER IN DESCRIPTIVE STATISTICS. LOOKING AT TABLE 1, AMONG THE ANALYTICAL SAMPLE IN THE CURRENT STUDY, 44.12% OF RESPONDENTS HOLD GRADUATE DEGREE. GIVEN THAT THE PERCENTAGE OF THOSE WHO HAVE BACHELORS OR HIGHER (NOT A GRADUATE DEGREE) IN MASSACHUSETTS (ONE OF THE STATES WITH THE HIGHEST PROPORTION OF COLLEGE GRADUATES IN THE U.S.) IS 44%, I SUSPECT THAT THIS IS TOO HIGH TO CONSIDER THE DATASET IS NATIONALLY REPRESENTATIVE. I APPRECIATE IT IF THE AUTHORS GO BACK TO THE DATA AGAIN TO CONFIRM WHETHER YOU DO NOT HAVE ANY CODING MISTAKES FOR EDUCATIONAL ATTAINMENT. RELEVANTLY, I COULD NOT FIND WHETHER THE AUTHORS USED APPROPRIATE WEIGHTS TO ACCOUNT FOR THE PROBABILITY TO PARTICIPATE IN THE SURVEY. BECAUSE UAS PROVIDES SAMPLING WEIGHTS, THEY SHOULD INCORPORATE THEM IN THE ANALYSES.

We appreciate your careful reading of the table. We checked the coding of the original variable and 44.7% of the respondents include those who hold bachelor’s degree, master’s degree, professional school degree (e.g., J.D., M.D.), and doctoral degree. Therefore, we clarified and corrected the categories in Table 1. Now the four categories are ‘less than high school,’ ‘high school,’ ‘some college,’ and ‘Bachelor’s degree or more’ (lines 141-142). According to the Census data, 38% of the U.S. population aged 25 and older have graduated from college. 44.7% is higher than the national average, and we have discussed this in the results section of the revised manuscript (lines 186-188). 

Also, in all analyses, UAS sampling weights were applied in order to produce nationally representative estimates. The sample weights were generated for each survey wave and account for unequal sampling probabilities of participants into the UAS survey (Kapetyn et al., 2020). We have included this in the methods section (lines 165-168).

2. THE METHODOLOGY IN THIS PAPER CANNOT MAKE CAUSAL CLAIMS, THOUGH THE GOAL OF THIS PAPER IS VERY CAUSAL (“WE HYPOTHESIZED THAT FOUR TYPES OF BEHAVIORAL ACTIVATION COPING … WOULD LOWER THE LEVELS OF DEPRESSIVE SYMPTOMS FOR THOSE LIVING ALONE”). ALTHOUGH I RECOGNIZED THAT THE AUTHORS TRIED TO ACCOUNT FOR TIME-VARIANT BY ADDING CONTROL VARIABLES, THIS IS NOT ENOUGH BECAUSE THERE COULD (AND SHOULD) BE UNOBSERVED CONFOUNDERS. THIS IS AN IMPORTANT LIMITATION IN THIS PAPER BECAUSE THE PAPER AIMS TO TEST WHETHER COPING STRATEGIES REDUCES DEPRESSIVE SYMPTOM AND THE AUTHORS PROVIDED POLICY IMPLICATIONS BY ASSUMING THAT THE OBSERVED CORRELATION IS CAUSATION. HERE IS ONE EXAMPLE SCENARIO THAT THE OBSERVED CORRELATION IS NOT CAUSATION. THE PAPER TRIED TO EXAMINE THE RELATIONSHIP BETWEEN LAGGED COPING STRATEGIES IN T-1 AND DEPRESSIVE SYMPTOM IN T; HOWEVER, THE AUTHORS DID NOT ACCOUNT FOR THE POSSIBILITY THAT THE LAGGED COPING STRATEGIES IN T-1 MAY BE AFFECTED BY DEPRESSIVE SYMPTOM IN T-2. THIS INDUCES A SELECTION ISSUE: LESS DEPRESSED INDIVIDUALS IN T-2 ARE MORE LIKELY TO EXERCISE IN T-1; THEREFORE, THE AUTHORS OBSERVED THE NEGATIVE ASSOCIATION BETWEEN EXERCISE IN T-1 AND DEPRESSION IN T. THERE ARE A LOT OF OTHER SCENARIOS THAT WOULD SHOW THAT THE ANALYTICAL STRATEGY IN THIS PAPER CANNOT MAKE CAUSAL CLAIMS. THE AUTHORS SHOULD DRAW DAG TO SHOW THEIR (ASSUMED) DATA GENERATING PROCESS, AND EXPLAIN WHY THEIR ASSUMPTION IS VALID. ALTERNATIVELY, THE AUTHORS MAY BE ABLE TO USE ESTABLISHED CAUSAL INFERENCE TECHNIQUES TO MAKE COMPELLING CAUSAL CLAIMS. OTHERWISE, I APPRECIATE IT IF THE AUTHORS EXTENSIVELY REVISE THE MANUSCRIPT TO AVOID CAUSAL LANGUAGES, INCLUDING THEIR HYPOTHESIS.

(ALSO RAISED IN COMMENT 11; COMMENT 12)

To clarify, the analytical technique employed in this study (xthybrid) does address endogeneity due to the omitted variables (i.e., unobserved measures) (Bell & Jones, 2015). However, as the reviewer pointed out, it does not deal with possible reverse causality between measures of depressive symptoms and coping strategies. Even though Stepanov and Suvorov (2017) developed a supplementary model to adjust for this possibility, there is no consensus on whether this model adequately accounts for reverse causality. Instead, we conducted sensitivity analysis to explore the possibility of reverse causality, as outlined in the paragraph below. 

We have addressed the reviewer’s comment in two different ways. First, regarding the concern of causality, we softened the causal language throughout the entire manuscript to not overstate our conclusions. Second, we re-estimated the hybrid model in order to examine whether lagged (t-1) depressive symptoms affected coping strategies in t. The results show that depressive symptoms at t-1 did not have significant effects on the coping strategies (exercise, meditation, relaxation, and social connection). This could be due to the fact that depressive symptoms are pretty stable throughout the study period except at the very beginning of the pandemic. We acknowledge that this does not rule out reverse causality completely, but it does indicate that behavioral activation coping strategies are associated with depressive symptoms and not vice versa during the study period. This is discussed in the revised manuscript (lines 296-302). 

3. THIRD, THE SAMPLE SIZE IS NOT CONSISTENT ACROSS THE MODEL. IN THE DESCRIPTIVE STATISTICS, THE SAMPLE SIZE IS 1,321, BUT THE ANALYTICAL SAMPLE IN TABLE 2 IS 1,280. THE AUTHORS SHOULD USE THE SAME ANALYTICAL SAMPLE.

In light of the reviewer’s comment, the sample size and descriptive statistics have been updated to reflect the final analytic sample size of 1,280 (Table 1). The results section was updated in order to reflect this change (lines 182-196).

4. FOURTH, I AM NOT SURE IF LISTWISE DELETION IS AN APPROPRIATE APPROACH TO TREAT THE MISSING VALUES IN INDEPENDENT VARIABLES, THOUGH THE NUMBER OF RESPONDENTS WITH MISSING VALUES IS SMALL. PROBABLY, IT IS WORTH TO TRY MULTIPLE-IMPUTATION FOR THE ROBUSTNESS CHECK. FURTHERMORE, THEY EXPLICATED THAT THEY DROPPED OBSERVATIONS IF RESPONDENTS DO NOT HAVE VALUES FOR COPING STRATEGIES, BUT I WOULD LIKE TO HEAR WHETHER THERE ARE NO MISSING VALUES IN THE OTHER INDEPENDENT VARIABLES.

We did not employ multiple imputation for multiple reasons. First, as the reviewer pointed out, the number of individuals with missing values are small (n=41). More importantly, these 41 individuals were not significantly different from the rest of the sample in terms of age, sex, depressive symptoms, education, income, chronic conditions, and substance use. Thus, we could assume that the cases were missing completely at random (MCAR) and, therefore, it was appropriate to employ listwise deletion technique (Kellermann, Travathan, & Kromrey, 2016). 

The UAS is an incredibly rich dataset without much missing data, particularly on demographic variables. For those living alone, the variable with the largest missing value was the current employment status with 4 individuals (0.68%). After eliminating cases with missing values for coping strategies, other covariates did not have any missing data. We have clarified our rationale for utilizing listwise deletion in the revised manuscript (lines 118-121).

5. FIFTH, I AM SOMEWHAT CONCERNED ABOUT THE VERY HIGH PHQ-4 SCORE IN WAVE I, BUT THERE WAS NO SUCH INCREASE AFTERWARD. I DO UNDERSTAND THAT THIS IS THE FIRST TIME THAT COVID-19 CASES HAVE DRASTICALLY INCREASED IN THE U.S., BUT WHY COULDN'T WE SEE A SIMILAR INCREASE AFTERWARD. IS IT EXPECTED? COULD YOU SEE A SIMILAR TREND FOR THOSE NOT LIVING ALONE? ARE THERE ANY STUDIES WHICH FOUND SIMILAR RESULTS?

This trend is very similar for those living alone and those living with cohabitants. Another study reports a trend that indicates an acute rise in PHQ-4 symptoms in late March and an abrupt drop immediately thereafter, to the month of June (Das, Singh, & Bruckner, 2022). This is briefly discussed in the revised results section (lines 191-192). 

6. FINALLY, I AM INTERESTED IN WHY THE AUTHORS INCLUDED TIME-INVARIANT VARIABLES EVEN THOUGH THEY INCORPORATED FIXED EFFECTS WHICH SHOULD ACCOUNT FOR TIME-INVARIANT EFFECTS.

It is correct that fixed effects account for measured and unmeasured time-invariant effects. However, we controlled for time-invariant variables in order to estimate random effects across waves and provide more robust parameter estimates for coping strategies. We conducted supplementary analyses with and without time-invariant covariates. Though including time-invariant covariates did not change the substantive conclusions, models with control variables yielded better fit indices (Table 2, Model 1 vs. Model 2 vs. Model 3). 

7. OVERALL, THE DISCUSSION SECTION SOUNDS FINE. I HAVE SEVERAL POINTS THAT I WOULD LIKE THE AUTHORS TO INCORPORATE. FIRST, I WOULD LIKE THE AUTHORS TO ADD HOW SUBSTANTIALLY IMPORTANT EXERCISE IS. I UNDERSTOOD THAT EXERCISE IS NEGATIVELY ASSOCIATED WITH DEPRESSIVE SYMPTOMS AMONG THOSE LIVING ALONE, BUT THE AUTHORS DID NOT PROVIDE INTERPRETATIONS FOR THE SIZE OF COEFFICIENTS. THIS DISCUSSION WILL STRENGTHEN THE ARGUMENT OF POLICY INTERVENTIONS.

We appreciate the reviewer’s positive comment and insightful feedback. The coefficients from xthybrid models are not presented as odds ratios, but they are standardized coefficients and can be interpreted as effect sizes, which we have done in the revised manuscript (lines 207-209). We also have discussed the policy implications of exercise in the context of the pandemic (lines 260-279). 

8. I ALSO WOULD LIKE THE AUTHORS TO RECONSIDER THE INTERPRETATIONS OF NON-SIGNIFICANT ASSOCIATIONS BETWEEN SOME COPING BEHAVIORS (I.E., MEDITATION, RELAXATION, AND SOCIAL CONNECTION) AND DEPRESSIVE SYMPTOMS. THE AUTHORS SPECULATED THAT INCREASED SEDENTARY BEHAVIORS DURING THE COVID-19 INCREASES THE IMPORTANCE OF EXERCISE, BUT THIS SHOULD NOT BE THE REASON WHY THE AUTHORS DID NOT SEE SIGNIFICANT ASSOCIATION BETWEEN THE OTHER COPING BEHAVIORS. SPECIFICALLY, I DO NOT THINK THAT SOCIAL CONNECTION IS NOT ASSOCIATED WITH DEPRESSIVE SYMPTOMS BECAUSE OF THE INCREASED SEATING TIME.

We agree with the reviewer that the significant findings for exercise do not mean the lack of practical significance for other coping strategies. We also did not intend to indicate that the lack of association between social connection and depressive symptoms was simply because of an increase in sedentary time. When we conducted a set of supplementary analyses in order to examine one coping strategy per analysis, we did not find any significant effects of meditation, relaxation, or social connection. This alleviates the possibility that the coefficient for exercise attenuated the effects of other coping strategies. 

Indeed, taking time to meditate, relax, and connect with loved ones has significant implications for psychological well-being. It is possible that other types of coping are effective for other mental health outcomes such as anxiety, eudemonic well-being, life satisfaction, or optimism. For example, frequent meditation and relaxation activate parasympathetic nervous system (e.g., slower heart rate). The revised discussion includes these possible explanations for the non-significant associations (lines 248-258). 

In particular, social connection is a complex concept encompassing the frequency of contact, frequency of interaction, mode of connection, physical contact, quality of interaction, and the object of social connection. For example, brief words of affirmation from a partner maybe more beneficial to psychological well-being than a two-hour conversation with an acquaintance. Thus, a single measure of social connection does not capture the complex tapestry of social relationships. We discuss the complexity of this variable (e.g., quality of social connections) within the context of our findings (lines 252-258) and as a limitation in the discussion (lines 312-317). 

9. ADDITIONALLY, I AM NOT SURE WHETHER WE CAN GENERALIZE THESE RESULTS TO ANOTHER LARGE-SCALE TRAUMATIC EVENT (“SECOND, THESE FINDINGS … A LARGE-SCALE TRAUMATIC EVENT”). BECAUSE COVID-19 PANDEMIC CARRIES UNIQUE FEATURES (LIKE SOCIAL DISTANCING), THIS MAY BE DIFFERENT FROM, FOR EXAMPLE, SEPTEMBER-11 ATTACKS. I APPRECIATE IT IF I CAN HEAR THE JUSTIFICATION WHY THE AUTHORS CAN CONCLUDE IN THIS WAY.

We agree with the reviewer that the COVID-19 pandemic poses unique challenges. The findings would have implications for potential large-scale public health threats that require a physical isolation measure. In light of this comment, we clarified this statement (line 274-275). 

10. FURTHER, I DID NOT UNDERSTAND THIS SENTENCE: “THE PRESENT STUDY SOUGHT TO UNDERSTAND HOW THE PANDEMIC INFLUENCED THE WELL-BEING OF ANOTHER HIGH-RISK GROUP: ADULTS LIVING ALONE.” THIS STUDY DOES NOT TEST THE EFFECT OF COVID-19 PANDEMIC ON DEPRESSIVE SYMPTOMS, BUT EXAMINED THE ASSOCIATION BETWEEN COPING STRATEGIES AND DEPRESSIVE SYMPTOMS DURING COVID-19 PANDEMIC.

In light of the reviewer’s comment, this statement was revised. “The present study sought to understand the well-being of another high-risk group during the pandemic: adults living alone” (lines 287-288).

11. LAST BUT NOT THE LEAST, GIVEN THE ABOVE-MENTIONED CAUSALITY ISSUE, I APPRECIATE IT IF THE AUTHORS AVOID CAUSAL LANGUAGES AND ADD THIS ISSUE AS A LIMITATION IN THE DISCUSSION SECTION.

(ALSO RAISED IN COMMENT 2; COMMENT 12)

As addressed in the response to Comment 2, we softened the causal language throughout the manuscript and addressed the concern in the limitation section (lines 296-302)

12. IN GENERAL, THE ARTICLE IS PRESENTED IN AN INTELLIGIBLE FASHION AND IS WRITTEN IN STANDARD ENGLISH. HOWEVER, AS NOTED IN THIS MEMO MULTIPLE TIMES, THE SUBMISSION IS MIXING CORRELATION AND CAUSATION. I COULD SEE MANY CAUSAL LANGUAGES, SUCH AS “EFFECT” AND “INFLUENCE,” IN THE MAIN TEXT. AGAIN, THE AUTHORS SHOULD AVOID THESE CAUSAL LANGUAGES UNLESS THEY CAN IDENTIFY CAUSAL EFFECTS BY USING ESTABLISHED CAUSAL INFERENCE TECHNIQUES.

(ALSO RAISED IN COMMENT 2; COMMENT 11)

As addressed in response to Comment 2 and 11, we have softened the causal language. Instead of words like ‘effect’ or ‘influence,’ we used terms like ‘association’ or ‘link.’ 

Please see our responses above for more detail. 

13. ANOTHER THING THAT I WAS CONCERNED ABOUT IS “GENERAL POPULATION” IN THE DISCUSSION SECTION. THE AUTHORS COMPARED THOSE LIVING ALONE TO THE GENERAL POPULATION, BUT I DO NOT UNDERSTAND WHAT THE “GENERAL POPULATION” MEANS. ARE THOSE NOT LIVING ALONE GENERAL POPULATION? IF SO, WHY CAN WE CONSIDER THAT THOSE LIVING ALONE ARE NOT “GENERAL POPULATION?”

“General population” means community dwelling adults with and without cohabitants. We clarified these statements (line 229). 

14. VERY MINOR POINTS: (1) PROBABLY, THE AUTHORS SHOULD AVOID ABBREVIATIONS, SUCH AS “SNAP ELIGIBILITY,” (2) PROBABLY, THE REFERENCE #26 IS WRONG BECAUSE I FOUND PHQ-15 AND PHQ-5 BUT NOT PHQ-4, (3) THE AUTHORS SHOULD USE A CONSISTENT LANGUAGE (E.G., COVID-19 PANDEMIC OR COVID-19 EPIDEMIC).

Per this comment, SNAP was unabbreviated (line 176). We updated the reference for PHQ-4 (line 134) and used the word ‘COVID-19 pandemic’ throughout the paper. 

References not included in the manuscript

Bell, A., Fairbrother, M. & Jones, K. (2019), Fixed and random effects models: making an informed choice. Quality and Quantity, 53, 1051-1074.

Kellermann, P. A., Travathan, D., & Kromrey, J. (2016). Missing Data and Complex Sample Surveys Using SAS®: The Impact of Listwise Deletion vs. Multiple Imputation on Point and Interval Estimates when Data are MCAR and MAR.

Stepanov, S. & Suvorov, A. (2017), Agency problem and ownership structure: outside blockholder as a signal. Journal of Economic Behavior and Organization, 133, 87-107.

---

## [Decision Letter · Decision Letter 1]

20 Apr 2022

Longitudinal Links between Behavioral Activation Coping Strategies and Depressive Symptoms of U.S. Adults Living Alone during the COVID-19 Pandemic

PONE-D-21-32992R1

Dear Dr. Kim,

We’re pleased to inform you that your manuscript has been judged scientifically suitable for publication and will be formally accepted for publication once it meets all outstanding technical requirements.

Kind regards,

Kenzie Latham-Mintus, PhD, FGSA

Academic Editor

PLOS ONE

Additional Editor Comments (optional):

Reviewers' comments:

Reviewer's Responses to Questions

**Comments to the Author**

1. If the authors have adequately addressed your comments raised in a previous round of review and you feel that this manuscript is now acceptable for publication, you may indicate that here to bypass the “Comments to the Author” section, enter your conflict of interest statement in the “Confidential to Editor” section, and submit your "Accept" recommendation.

Reviewer #1: All comments have been addressed

2. Is the manuscript technically sound, and do the data support the conclusions?

Reviewer #1: Yes

3. Has the statistical analysis been performed appropriately and rigorously? 

Reviewer #1: Yes

4. Have the authors made all data underlying the findings in their manuscript fully available?

Reviewer #1: Yes

5. Is the manuscript presented in an intelligible fashion and written in standard English?

Reviewer #1: Yes

6. Review Comments to the Author

Reviewer #1: (No Response)

7. PLOS authors have the option to publish the peer review history of their article (what does this mean?). If published, this will include your full peer review and any attached files.

Reviewer #1: No

---

## [Editor Report · Acceptance letter]

25 Apr 2022

PONE-D-21-32992R1 

Longitudinal Links between Behavioral Activation Coping Strategies and Depressive Symptoms of U.S. Adults Living Alone during the COVID-19 Pandemic 

Dear Dr. Kim:

I'm pleased to inform you that your manuscript has been deemed suitable for publication in PLOS ONE. Congratulations! Your manuscript is now with our production department. 

Kind regards, 

on behalf of

Dr. Kenzie Latham-Mintus 

Academic Editor

PLOS ONE